

# Distributed observations of wind direction using microstructures attached to actively heated fiber-optic cables

Karl Lapo[1,2], Anita Freundorfer[1], Lena Pfister[1], Johann Schneider[1], John Selker[3], and Christoph Thomas[1,2]

[1]University of Bayreuth, Bayreuth, Germany
[2]Bayreuth Center of Ecology and Environmental Research, Bayreuth, Germany
[3]Department of Biological and Ecological Engineering, Oregon State University, Corvallis, Oregon, USA

**Correspondence:** Karl Lapo (karl.lapo@uni-bayreuth.de)

**Abstract.** The weak-wind boundary layer is characterized by turbulent and submeso-scale motions that break the assumptions necessary for using traditional eddy covariance observations such as horizontal homogeneity and stationarity, motivating the need for an observational system that allows spatially resolving measurements of atmospheric flows near the surface. Fiber-Optic Distributed Sensing (FODS) potentially opens the door to observing a wide-range of atmospheric processes on a spatially

distributed basis and to date has been used to resolve the turbulent fields of air temperature and wind speed on scales of second and decimeters. Here we report on progress developing a FODS technique for observing spatially distributed wind direction. We affixed microstructures shaped as cones to actively-heated fiber-optic cables with opposing orientations to impose directionally-sensitive convective heat fluxes from the fiber-optic cable to the air, leading to a difference in sensed temperature that depends on the wind direction. We demonstrate the behavior of a range of microstructure parameters including aspect ratio,

spacing, and size and develop a simple deterministic model to explain the temperature differences as a function of wind speed. The mechanism behind the directionally-sensitive heat loss is explored using Computational Fluid Dynamics simulations and infrared images of the cone-fiber system. While the results presented here are only relevant for observing wind direction along one dimension it is an important step towards the ultimate goal of a full three-dimensional, distributed flow sensor.

## 1   Introduction

Laser pulses sent along a fiber optic cable scatter back along the path of the fiber with a temperature dependent shift in frequency, providing a powerful geophysical sensing technique called Distributed Temperature Sensing (DTS) (Selker et al., 2006; Tyler et al., 2009). The principle behind DTS has been used to observe a wide range of geophysical processes and aerial deployments of DTS are a promising avenue for observing atmospheric processes on a distributed basis (Pfister et al., 2016; Thomas et al., 2012; Zeeman et al., 2015). Previous work with atmospheric DTS has demonstrated the ability to observe atmo-

spheric temperatures (Thomas et al., 2012), wet bulb temperature (Euser et al., 2014; Schilperoort et al., 2018), solar radiation (Sigmund et al., 2017), and wind speed (Sayde et al., 2015) at a fine spatial and temporal resolution. We refer to this broader application of DTS technology as Fiber Optic Distributed Sensing (FODS).

FODS has the potential to fill the missing scales between point observations and remote sensing. FODS can provide observa-





tions of atmospheric variables at temporal resolution as fine as one second and spatially distributed observations on scales from tens of centimeters to kilometers. In particular, FODS is ideally suited for observing turbulence, especially during weak-wind conditions. Weak-wind boundary layers break many of the assumptions that underlie eddy covariance techniques (Thomas, 2011; Cheng et al., 2017; Pfister et al., 2019), which forms an obstacle for understanding the dynamics of turbulence during

these condition. For instance, eddy covariance relies on the ergodic hypothesis, the assumption that time and space averages converge under horizontally homogeneous and stationary conditions (Taylor, 1938). From this assumption, the time-averaged flow can be used to infer the spatially-averaged fluxes and turbulent properties. However, weak-wind boundary layers break this critical assumption, thereby limiting the inferences we can make about the nature of turbulence from point observations alone, even within exceptionally dense observation networks (Pfister et al., 2019; Mahrt et al., 2009). Further, weak-wind boundary

layers violate the assumptions behind similarity theory, with non-local and intermittent fluxes (Sun et al., 2012, 2015), creating substantial problems for climate, weather, and land models which rely on similarity theory to simulate turbulent fluxes at the land surface (Holtslag et al., 2013).

The ability to observe spatially-distributed wind direction, in addition to wind speed and temperature, at a fine spatial and temporal scale near the surface, would be a powerful technique for studying atmospheric turbulence. However, prior work

has only been able to observe the magnitude of wind speed normal to the fiber, not the direction (Pfister et al., 2019; Sayde et al., 2015; Ramshorst et al., 2019). The approach for observing flow direction with FODS explored in this study is based upon the hypothesis that microstructures with opposite orientations placed on paired, actively-heated fiber-optic cables impart a directionally-sensitive convective heat loss (section 2.1).

Here, we present results from a series of wind tunnel experiments that demonstrate the basic feasibility of observing wind

direction with FODS. Additionally, an empirical expression for describing the FODs signal of wind direction was developed (section 2.1), parameters that govern the magnitude of the signal were tested (section 3.2), the uncertainty and scale of the wind direction signal were evaluated (section 3.3), and the mechanism behind the wind direction signal was verified (section 3.4). Finally, these results are discussed within the context of the remaining challenges for observing spatially-distributed wind direction in an environmental application (section 3.5).


## 2 Methods

### 2.1 Motivating the microstructure approach

Raman spectra DTS operates on the principle of temperature-dependent backscattering of photons at a higher and lower frequency than the original laser pulse. The reader is referred to Selker et al. (2006) and Tyler et al. (2009) for a detailed description

of the operating principal. This temperature dependency can be used to observe air temperature directly. Wind magnitude orthogonal to the cable can be observed using the temperature difference between an active, resistively-heated cable and a paired, unheated cable, similar to the principle of hot-wire anemometery (Sayde et al., 2015; Ramshorst et al., 2019). Actively heating the cable causes it to be warmer than the atmosphere, thus the convective heat flux cools the heated cable: stronger winds cause





a larger, cooling convective heat flux and a smaller temperature difference between the paired cables (Sayde et al., 2015).

To observe wind direction, we propose a similar approach combining the active heating with microstructures printed on the

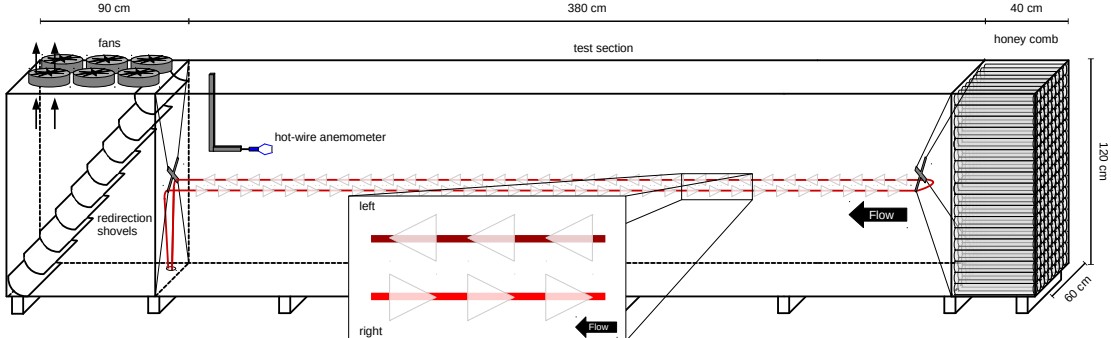

**Figure 1.** Schematic of the low-speed wind tunnel used to test the microstructure approach to detecting wind direction. The subset highlights the cone orientation relative to the mean flow of the tunnel. The FOC with left pointing cones is cooler than the FOC with the right pointing cones.

the FOC. The underlying assumption is that applying asymmetric microstructures with opposite orientations to paired, actively heated cables induces directional differences in the turbulent flow around the microstuctures and thus in the convective heat

loss from the FOC to the air. This difference in convective heat loss results in a temperature difference between the two cables that can be sensed by FODS (see Figure 1).

The convective heat loss from a surface with roughness elements can be written as (Owen and Thomson, 1962)

$$Q_h = \rho u_* c_p (T_a - T_s) \alpha^{-1} Re_*^{-m} Pr^{-n} \tag{1}$$

where $\rho$ is the density of air, $c_p$ is the specific heat of dry air, $T_a$ is the temperature of the air, $T_s$ is the temperature of the fiber, $\alpha$ is an empirical constant related to the roughness of the surface, Pr is the Prandtl number. $m$ and $n$ are empirical constants.

$Re_*$ is the roughness Reynolds length defined as

$$Re_* = \frac{u_* h}{\nu} \tag{2}$$

where $u*$ is the friction velocity, $h$ is the "sand-equivalent height" of the roughness elements or the thickness of the surface layer (Owen and Thomson, 1962), and $\nu$ is the kinematic viscosity. Combining equations 1 and 2 yields

$$Q_h = \rho u_*^{1-m} c_p (T_a - T_s) \alpha^{-1} \left(\frac{h}{\nu}\right)^{-m} Pr^{-n} \tag{3}$$

Cones pointing into the flow should have a lower equivalent roughness height than cones pointing with the flow as a result of changing from a streamlined shape into a bluff shape. Consequently, actively heated cables with microstructures of opposite

orientations should have different cooling convective heat fluxes and resulting temperatures.

The difference in the convective heat flux along each cable is manifested as a temperature difference between the cables from





which we define the wind direction signal:

$$\Delta T = T_{right} - T_{left} \tag{4}$$

Cables with microstructures pointing into the flow (Fig. 1) should have a smaller (cooling) convective heat flux and higher temperature ($T_{right}$) compared to cables with microstructures pointing with the flow, which should have a larger convective heat flux and resulting lower temperature ($T_{left}$). In the displayed case (Figure 1) $\Delta T > 0$ would indicate a wind direction from the right, while $\Delta T < 0$ indicates a wind direction from the left.

To derive a functional form for $\Delta T$ as a function of wind speed, we substitue subtract $T_{right}$ and $T_{left}$ into equation 1 and subtract the two quantities.

$$Q_{left} - Q_{right} = \rho c_p \left(\frac{h}{\nu}\right)^{-m} u_*^{(1-m)} \left(\frac{(T_a - T_{right})}{\alpha_{right}} - \frac{(T_a - T_{left})}{\alpha_{left}}\right) \tag{5}$$

Solving the equation for $T_{left} - T_{right}$ requires the assumption that $\alpha_{right} \ll \alpha_{left}$, i.e. that the right pointing cones have a much smaller effective roughness than left pointing cones, equation 5 reduces to

$$T_{left} - T_{right} = \frac{Q_{left} - Q_{right}}{\rho c_p \left(\frac{h}{\nu}\right)^{-m}} \alpha_{left} \alpha_{right} u_*^{m-1} \tag{6}$$

Equation 6 suggests that the relationship between $\Delta T$ and wind speed, $U$, should be a power law with a negative exponent, as $m$ should be less than 0.5 and as $u_*$ and $U$ are linearly related quantities (Stull (1994)). Equation 6 possesses the desired behavior that as the wind speed increases, the difference in fiber temperature should also converge towards zero. The relationship between $\Delta T$ and $U$ in equation 6 relies on several assumptions so an exponential decay model was also employed as a competing hypothesis as some sort of non-linear, decreasing relationship between $\Delta T$ and $U$ is anticipated.

## 2.2 Instruments and Wind Tunnel

The microstructure approach to distributed observations of wind direction was tested in a controlled wind tunnel environment. The wind tunnel was designed to provide small turbulence intensity for low velocity flows. The suck-through wind tunnel test section was 3.0m long, 0.6m wide, and 1.2m tall. At the entrance of the wind tunnel the flow was straightened and external turbulence was combed out using a honeycomb section made from 0.20m long pipes with a 0.05m radius. The flow was allowed to settle over 0.6m before entering the test section. The tunnel was lined with 0.03m thick insulating polyurethane boards to eliminate spatial differences in surface temperature and thus provided a coherent radiative environment within the tunnel, which minimizes differences in the longwave radiative transfer that arises from even subtle differences in surface temperature and emissivity. Deviations of longwave radiation parallel to the cable can cause artifacts in the DTS.

A sonic anemometer (Model CSAT3, Campbell Scientific Inc., Logan, UT, USA) was used characterize the flow within the wind tunnel. The tunnel was controlled to yield consistent wind speeds for each test with along-flow velocities as given in Table 1. Each parameter was tested with ten minutes of observations using a flow that did not vary in time. The center of the wind





tunnel was free from the influence of a shallow wall boundary layer. At the back of the tunnel there was some enhancement of turbulent mixing due to deflection towards the fans, which was excluded from further analysis. Horizontal turbulent intensity varied between 0.005 and 0.01 for the lowest to highest wind tunnel velocities. The friction velocity along the central axis of the wind tunnel ranged from $0.0025\,\frac{m}{s}$ for the lowest wind speeds to $0.02\frac{m}{s}$ for the highest wind speeds. The turbulence within

the tunnel was substantially lower compared to atmospheric flows and may impact direct transfer of wind tunnel results to field conditions.

The time-averaged wind speed during tests with the FOC was monitored using a one-dimensional hot-wire anemometer (Model, TA300, Trotec, Heinsberg, Germany) with a precision of $\pm0.2ms^{-1}$. To provide an independent measurement of the fiber temperatures a high-resolution thermal infrared camera was used to record its brightness temperature (Model PI640, Optris,

Gerlin, Germany). The camera observes the wavelengths of 7.5 to $13\mu m$ with a pixel resolution of 640x480 and an accuracy of $\pm2^{\circ}C$. The camera was placed on the floor of the wind tunnel looking up at the FOCs. Pictures of the fiber and microstructure brightness temperatures were acquired at 1 Hz and averaged over 10s. In order to avoid angular effects of the emitted thermal radiation, the cones and fiber were coated with infrared paint (Washable thermographic paint for special applicationss, LabIR, Pilsen, Czech Republic) that has a known emissivity of 0.94 to 0.97 for viewing angles $60^{\circ}$ to $5^{\circ}$, respectively.

## 2.3   Fiber-optic array

One continuous FOC was deployed in the test section, parallel to the flow within the wind tunnel (Fig. 1). Microstructures were mounted in opposing directions on the other pair of heated fibers (zoomed in region Fig. 1). The cables were mounted within the tunnel using square aluminum crosses with a width of 7.5cm and were gently looped around the back of the crosses to avoid sharp bends which cause a signal loss (Selker et al., 2006). The FOC had a 0.82mm stainless steel sheath with a 0.15mm PVC

coating, yielding a total outer diameter of 1.12mm and the actual fiber optic cable was loosely buffered and gel-filled (Model C-Tube, Brugg, Brugg, Switzerland). The cable was heated electrically by applying a current to the high-resistance (2.3 $\Omega$/m) stainless steel sheath.

Fiber temperature was observed using a high-resolution DTS instrument (Model 5km Ultima, Silixa, London, United King-

don). The DTS has a temperature resolution of $0.01^{\circ}C$, although this value is dependent upon the temporal averaging used. The spatial resolution is 0.127m with a temporal resolution of $\approx1s$. The intensity of the back-scattered light was converted to a temperature using calibrated parameters that vary with instrument temperature and fiber properties. We explicitly calculate these parameters through a matrix inversion of the back-scatter equation using three reference sections (Hausner et al., 2011). This calibration technique eliminates effects from differential attenuation and instrument properties that can vary with time.

The reference sections are composed of warm and cold calibration baths. The cables were deployed in the calibration baths both prior to entering the wind tunnel and after the fiber exits the tunnel, yielding two temperatures at four locations along the fiber. Three of these calibration sections were used to solve for the calibrated parameters with the fourth withheld for characterizing the instrument uncertainty. Each reference bath was well-mixed using aquarium pumps to avoid stratification. The fiber was loosely coiled within the baths such that they did not contact the bath walls. Two class-A PT100s, with an accuracy of $0.15^{\circ}C$,





**Table 1.** Tested parameters within this study.

| Variable | Values |
| --- | --- |
| Cone size | 0.012 m, 0.016 m |
| Cone spacing | 0.02 m, 0.05 m, 0.1 m |
| Cone aspect ratio | 1:1 (regular), 1:2 (long and skinny), 2:1 (short and wide) |
| Heating rate | $0.5Wm^{-1}$, $1.5Wm^{-1}$, $2.5Wm^{-1}$ |
| Mean wind speed | $0.35ms^{-s}$, $0.9ms^{-s}$, $1.8ms^{-s}$, $3.8ms^{-s}$ |

were deployed in each calibration bath. After calibration, the DTS had an root mean square error of $0.61°$C (n=4300000) when evaluated against the temperature of the reference bath, in line with the published accuracy form the manufacturer. We used this error as an estimate of the instrument uncertainty (section 3.3).

### 2.3.1 Heating

The heating of the FOC was provided by a high-precision heating unit (Model Heat Pulse System, Silixa, London, United Kingdom) which applies a known heating rate per section of cable. Multiple heating rates were tested (Table 1) as previous work has suggested that heating rate can influence the accuracy of FODS of wind variables (Sayde et al., 2015) as the convective heat loss is a linear function of the temperature difference between the cable and air temperatures.

### 2.3.2 Microstructures

We used 3D-printed (Model Form 2, formlabs, Berlin, Germany) cones that can be affixed to the FOC. Cone base width was selected as 12mm and 16mm, with the ratio of width:height varying from 1:1 (as tall as it is long), 1:2 (long and skinny), and 2:1 (short and wide; see Table 1). These cones were then affixed to the paired, heated cables with each cable having cones oriented in the opposite direction from the other, as shown in Figure 1. The distance between cones was varied between 2cm

and 10cm (Table 1). While we solely used cones for this study, we speculate that additional shapes may be used to achieve a similar directional dependence.

### 2.4 Numerical Simulations

Computational Fluid Dynamics (CFD) simulations were completed to inform initial design decisions of the microstructures and to verify the observed heat transfer mechanism (section 3.4). We used the OpenFOAM Computational Fluid Dynamics software

(openfoam.com) with the simpleFoam solver and the standard k-$\epsilon$ turbulence model for doing a 3-dimensional simulation of the flow along a fiber with microstructures. The simulations were done using a long enough piece of fiber such that the flow could adjust to the microstructures. We tested the heating rate, microstructure size, aspect ratio, and spacing each at a variety of wind speeds. The initial CFD simulations allowed the targeting of a specific range of variables.





## 3 Results and Discussion

### 3.1 Temperature differences

The results confirmed our initial assumption of directionally sensitive heat loss, and thus cable temperatures, from the cones
pointing in different directions (see Fig. 2 for an example for a single test of cone spacing, size, aspect ratio, and heating rate
as a function of wind speed). The largest temperature difference between cables coincides with the location of the microstruc-
tures (between 0.75m and 2.25m). At the beginning (x=0m) and end (x=3m) of the test section the cable temperature exhibits
artifacts caused by the support crosses that are used to mount the fiber in the tunnel. The microstructure fibers exhibit a uniform
temperature within the test section except at the lowest wind speed, in which a decrease in temperature is observed with length
along the tunnel, perhaps as a result of unorganized turbulence within the tunnel at these low wind speeds. The heated fibers
cool as the wind speed is increased, as expected.

At all wind speeds in the region with cones, the fiber with cones pointing left (Figure 1) has a lower temperature than the
fiber with cones pointing right. $\Delta T$ is largest at the lowest wind speeds and becomes small enough at the highest wind speed
that the temporal variability in temperature for the fibers overlap (shaded regions in Figure 2d). The effect of the uncertainty
in the $\delta T$ signal is discussed further in section 3.3. The reduction in the $\Delta T$ signal with higher wind speeds may be caused
by the enhanced roughness from both microstructure orientations being unable to increase the sensible heat flux beyond some
maximum value.

When determining the temperature difference, we examine the test section in which the cone signal is not affected by edge
effects (0.75m to 2.25m along the tunnel). The temperature signal for both fibers is linearly interpolated to a common x-
coordinate along the tunnel.

### 3.2 Optimizing the microstructure configuration

The temperature signal is defined as the mean difference, in both time and space, according to Eq 4. A positive $\Delta T$ is expected,
as the fiber with left pointing cones should be warmer than the fiber with the right pointing cones. The tested parameters were
stratified from the most influential (Figure 3a) to least influential (Figure 3c) factors. A hyperbolic and an exponential model
were fit to the set of best performing parameters (Figure 3d).

The most influential parameter was the cable heating rate. A heating rate between $1.5Wm^{-1}$ and $2.5Wm^{-1}$ yields the
largest $\Delta T$ with a mean difference of approximately equal to 1.0K at the lowest wind speed and 0.3K at the highest wind speed
(Figure 3a). Certain combinations of microstructure properties with higher heating rates have a smaller $\Delta T$ than different
combinations of microstructure properties with a lower heating rate. However, a larger heating rate leads to a larger $\Delta T$ for a
fixed combination of cone size, spacing, and aspect ratio.
The cone spacing was the second most influential factor, largely due to the smallest cone spacing of 0.02 m (Figure 3b). The

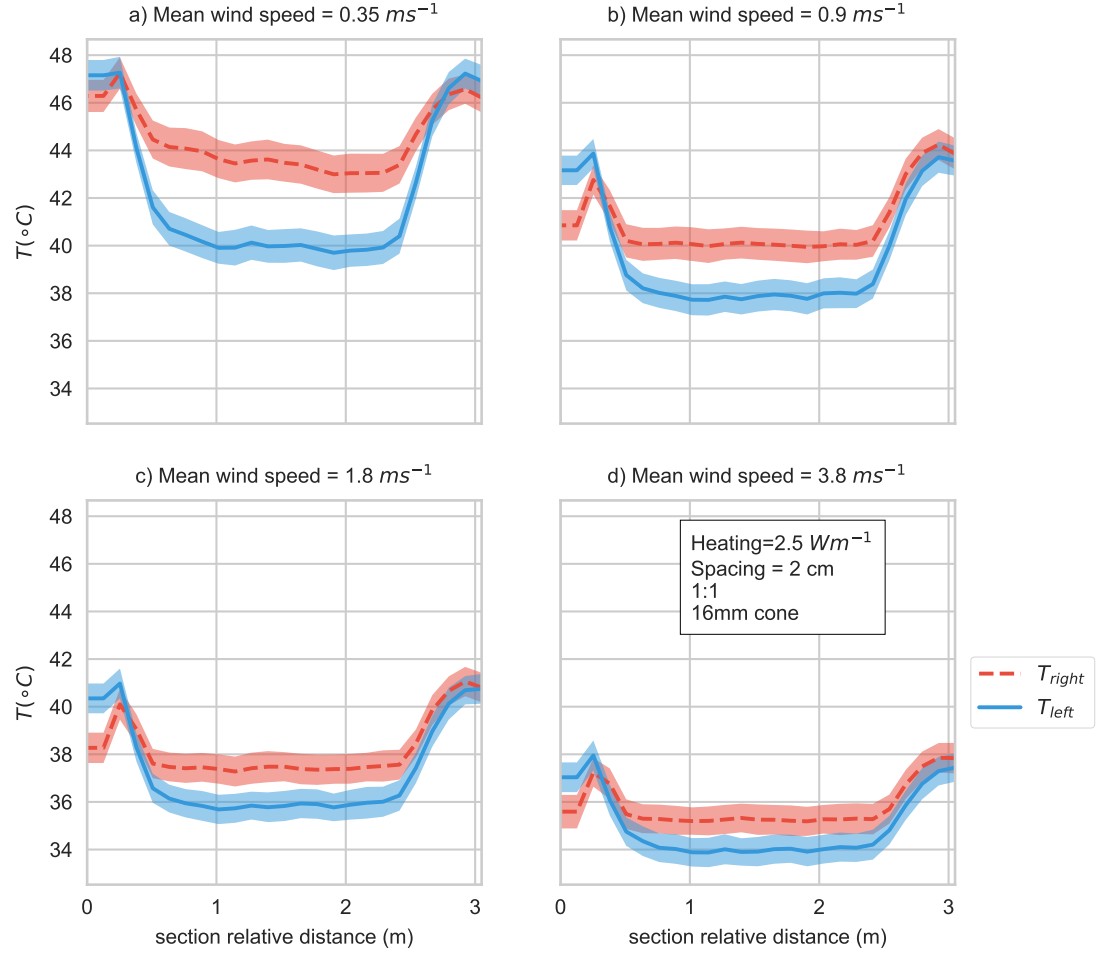

**Figure 2.** Time-averaged temperature along the test section in the wind tunnel for a horizontal wind speed of (a) $0.35ms^{-1}$, (b) $0.9ms^{-1}$, (c) $1.8ms^{-1}$, (d) $3.8ms^{-1}$ for the parameters listed in the Figure. The standard deviation of temperature in time for each point along the FOC is shown in the filled colors. The cones are only present between 0.75m and 2.25 of the tunnel section. The mean wind direction is in the positive x-direction.

mean $\Delta T$ for the 0.05 m and 0.1 m spacing were indistinguishable at higher wind speeds, while the 0.02 m spacing resulted in the highest $\Delta T$. The other parameter values tested, such as cone aspect ratio and size, had a limited effect (Figure 3c). The exception was the aspect ratio, width:height, of 1:2, which led to the smallest $\Delta T$ as the other two aspect ratios converged to the same $\Delta T$ for higher wind speeds. At lower wind speeds, the effect of cone size and aspect had an inconsistent effect. For instance, the 0.016 m cones had the largest temperature difference for the 1:1 cones and the smallest difference for the 2:1 cones. This ordering reversed itself at higher wind speeds (Fig. 3c). However, these differences were well within the observational uncertainty for the DTS device and should not be interpreted. From this we conclude that any combination of the 0.012 m and



**Figure 3.** The $\Delta T$ for all combinations of cones and heating rates (see Table 1). The data are stratified using the variable with the largest impact, i.e., temperature difference between coned cable sections pointing in different directions. (a) All temperature differences as a function of wind speed classified by heating rate. (b) The highest heating rate is selected (red points in (a)) and data are classified according to the spacing between cones. Data with lower heating rates are shown in grey. (c) The 2cm spacing and highest heating rates (red points in (a) and (b)) are selected and then classified according to the cone aspect ratio. The 12mm cones are marked with pluses and the 16mm cones are marked with right pointing triangles. All other heating rates and cone spacings are shown in grey. (d) The best performing parameters are fit with an exponential and power law model (see section 2.1 for details).





0.016 m cone sizes and the 1:1 and 2:1 aspect ratios were appropriate for developing the distributed wind direction observation system.

We further evaluated whether $\Delta T$ is well-described by a power law or exponential decay relationship with wind speed. The power law function outperformed the exponential decay as it has smaller residuals and a lower uncertainty in the fit parameters. Additionally, it was able to describe all sets of tested parameters as a function of wind speed, albeit with different parameter values, while the exponential fit cannot (not shown). The results here suggest that the basic relationship shown in equation 6 is applicable. It should be noted that both functions have problematic limiting behavior as the wind speed approaches zero. Further work will be necessary to identify a minimum wind speed threshold at which $\Delta T$ becomes significantly non-zero.

### 3.3 Certainty in estimating the wind direction signal

The wind direction signal, $\Delta T$ is subject to a non-negligible uncertainty from the DTS device. The uncertainty for $\Delta T$ is formulated as

$$\delta_T = (2\sigma_{DTS}^2)^{\frac{1}{2}} \tag{7}$$

where $\delta_T$ is the uncertainty in the $\Delta T$ signal and $\sigma_{DTS}$ is the mean standard error derived from the DTS reference thermometers immersed in the calibration baths of uniform temperature. For the DTS device used in this study, the mean standard error is $0.61°C$ (section 2.3) yielding an uncertainty $\delta_T = 0.81°C$. From this it follows that any $\Delta T \leq \delta_T$ cannot be distinguished from noise.

The distribution of $\Delta T$ in both time and space (n=20700) for a given experiment was normally distributed around the mean difference (Figure 4a,b). The strength of the wind direction signal was inversely related to wind speed (Figure 2) and as a consequence the fraction of $\Delta T \leq \delta$ increased with the wind speed. At the highest wind speed tested, some $\Delta T$ values even changed sign, which would result in an incorrect wind direction estimation(Fig. 4b).

One strategy for improving the accuracy of the wind direction signal is to average the temperature signal from both cables in time and space prior to computing $\Delta T$. A running average of temperature was calculated using a variable number of time (1s) and/or space (0.127m) observations (Figure 4c,d). Averaging the temperature signals over longer temporal and spatial intervals than the native resolution of the DTS device reduced the fraction of $\Delta T$ signal below the instrument accuracy and hence increased the fraction of observations suitable for wind direction determination (Figure 4c). For a given set of microstructure parameters, all wind speeds benefit from averaging the FODS signals. The exception is the lowest wind speed of $0.35ms^{-1}$, which effectively always yielded a $\Delta T$ larger than $\delta_T$ (Figure 4c). In this case, the averaging reduces the temporal resolution of the wind detection calculations. All acceptable cone size and aspect ratios found in section 3.2 had similar responses to averaging the FODS signal except the 0.012 m cones with a 2:1 aspect ratio yielded slightly greater uncertainty than other combinations (Figure 4d).

Any averaging reduces the resolution of the method. Fewer time intervals are necessary for improving the certainty in the $\Delta T$ signal compared to spatial averages for all wind speeds (Figure 4c). However, an interpretation solely based upon number of



**Figure 4.** The distribution of temperature differences between cables with opposing cone orientations for the 16mm, 1:1 cones with 2cm spacing and a heating rate of $2.5 W m^{-1}$ for (a) a wind speed of $0.35 m s^{-1}$ and (b) a wind speed of $3.8 m s^{-1}$. The grey region indicates the uncertainty, $\delta_T$. The percentage of observations with $\Delta T < \delta_T$ is indicated. (c) A two-dimensional histogram of the percentage of $\Delta T < \delta_T$ as a function of averaging interval in time and space for the highest wind speed (same data as in b). The contours for the 1% and 5% thresholds are indicated for each wind speed. Note that the $0.35 m s^{-1}$ wind speed is always below the 1% threshold. (d) The 1% and 5% contours for the acceptable cone aspect ratios and sizes at the highest wind speed.





averaging intervals may be misleading as the wind direction method is aimed at observing atmospheric turbulence, especially for the weak-wind boundary layer featuring short-lived and small-scale motions. Instead, we seek to find spatial and temporal averages that facilitate the observation of mean direction for a typical eddy length scale during those conditions. Taylor's hypothesis (Taylor, 1938) is commonly employed for estimating the eddy scale that can be resolved with a particular instru-

ment resolution. With Taylor's hypothesis time scales can be converted into spatial scales by assuming a relationship between temporal and spatial gradients through

$$\ell = U\tau \tag{8}$$

where $U$ is the mean wind speed, $\ell$ is the spatial scale of a turbulent eddy, and $\tau$ is the time scale. At a wind speed of $4ms^{-1}$, a $4s$ average results in observing eddies of length scale of $\approx 16m$. This eddy length scale is larger than those commonly found in in the weak-wind boundary layer. Whereas a spatial average with n=10 observations equates to an observation every 1.27m.

Using the DTS temporal resolution, for the same $4ms^{-1}$ wind speed, an eddy of length scale $\approx 4m$ can be observed. For this reason, improving the certainty in $\Delta T$ at higher wind speeds through spatial averaging is recommended.

### 3.4   Explaining the physical mechanism for directional heat loss

To provide an independent verification of the FODS signals and investigate the mechanism behind the observational principle, a thermal infrared camera was employed to observe the fiber and cone brightness temperatures (Figure 5a-c). The use of a paint

with a relatively constant emissivity with viewing angle allows a comparison of brightness temperature between different parts of the microstructure fiber-optic cable setup. The brightness temperature revealed the fine spatial structure of temperatures across the microstructure-cable, from which we inferred differences in the convective heat loss. The brightness temperature for both the left and right pointing cones were cropped from the larger image (Figure 5ab) using a threshold value. The difference in temperature between the left and right pointing fibers were driven by differences in temperature in two locations. The base

of the cones were the coolest part of the cable/cone system with the base of the left pointing cones being $\approx 2°C$ cooler than the base of the right pointing cones (Figure 5c). The cable immediately behind the right pointing cones, in the lee of the flow, was the warmest point along the fiber and was warmer than the equivalent segment on the left pointing cones by $> 2.5°C$. This difference in temperature decreased with length along the cable towards the next pair of cones.

The brightness temperatures suggested that two factors explain the directional sensitivity of the heat loss to the microstruc-

tures. First, the low temperatures on the base of the left pointing cones imply an enhancement of turbulent exchange at the base of the cone relative to the right pointing cones. Secondly, the high fiber temperatures behind the right pointing cones imply that the right pointing cones are sheltering the cable in their lee, reducing the cooling by limiting the convective heat flux. The sheltering effect may partially explain why our findings for the cone spacings of 0.05 m and 0.1 m were indistinguishable (Figure 3b), as the sheltering occurs over a distance of approximately 0.01-0.02 m. As the spacing increases past the range in

which the sheltering occurs, the fibers with right and left pointing cones converge to the same temperature (not shown).

The two turbulence features suggested in the brightness temperatures are further demonstrated using CFD simulations for the same fiber set-up shown in Figure 5ab. The simulated turbulent kinetic energy (TKE) is used as a proxy for the convective heat



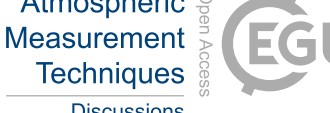

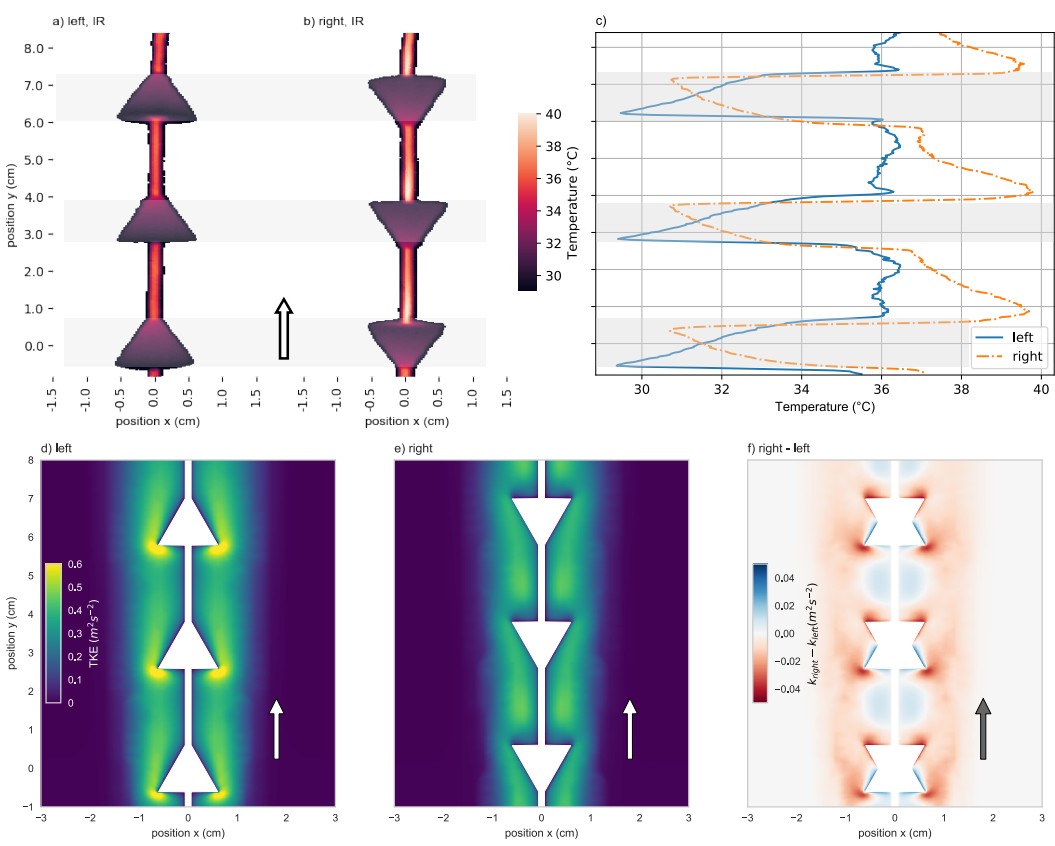

**Figure 5.** The brightness temperature of the fiber-optic cables with (a) left and (b) right pointing cones for the 12mm, 1:1 cones with 2cm spacing and a heating rate of $2.5Wm^{-1}$ for a wind speed of $0.9ms^{-1}$. (c) The average of the three warmest pixels in each horizontal row is provided to better demonstrate the spatial pattern of temperatures along the fiber. The grey shading is to visually line up the cones in a-c. (d, e) Turbulence kinetic energy (TKE) computed from the CFD simulations for the same experiment as in (a) and (b) with (f) the difference in TKE between the two orientations. The difference in TKE was not physically meaningful in the region with a cone in either (d) or (e) and is excluded in (f). The subset arrows indicate the direction of the mean flow.



exchange. TKE is defined as

$$TKE = \frac{1}{2}\left(\overline{(u')^2} + \overline{(v')^2} + \overline{(w')^2}\right) \tag{9}$$

where $u$, $v$, and $w$ are the three orthogonal wind velocity components and the $'$ denotes the temporal perturbation from Reynold's averaging. The differences in TKE corresponded to the features found in the brightness temperature. The left pointing cones substantially enhanced the turbulent exchange at the base of the cones compared to the right pointing cones (Figure

5f). Additionally, the right cones strongly reduced TKE in the lee of the cones (Figure 5f). Both effects caused the left pointing cones to being subject to enhanced convective heat exchange relative to the right pointing cones. The right pointing cones did provide a small enhancement of turbulent exchange further along the fiber, past the region in the lee of the cone (blue colors in Figure 5f). This enhanced turbulent exchange was also apparent as the brightness temperature of the right pointing cones decreased from the base of one cone to the tip of the next one (Figure 5c). However, this small increase in turbulent exchange

was not large enough to overcome the reduction in turbulent exchange directly in the lee of the cones. Both the CFD modeling approach and the independent experimental brightness temperatures confirmed that the wind direction signal from the oriented microstructures results from differences in the convective heat exchange generated by the microstructures.

### 3.5    Remaining questions and future work

This study only demonstrated the ability to observe wind direction within one dimension. Convolving the FODS wind direction

observations along orthogonal directions into a fully three dimensional wind field is a substantial challenge and beyond the scope of this proof-of-concept study. The angle of attack of the mean wind direction along the fiber will influence of the wind direction signal, similar to issues with observing wind speed with FODS (Sayde et al., 2015; Pfister et al., 2019; Ramshorst et al., 2019). Exploring the effect of wind attack angle was not possible given the size and aspect ratio of the wind tunnel used in this study. The ability to detect mean wind direction is useful, but developing a flow sensor for studies of atmospheric

turbulence also requires the ability to detect rapidly changing wind vectors. The cone/cable system has some thermal inertia that creates a lag in reaching an equilibrium $\Delta T$. This study only highlights the ability to measure time- and space-averaged flow, which may not be sufficient to resolve the energy-containing eddies for weak-wind boundary layers. Additional work is necessary for determining the time response of the wind direction signal. Finally, the flow explored in this proof-of-concept study has a lower turbulence intensity than atmospheric flows. Deployments in real atmospheric flows may require a larger

heating rate or further averaging in order to observe a meaningful $\Delta T$. These considerations need to guide future work to enable true three-dimensional observations of wind speed and direction.

The method for detecting wind direction depends on the temperature difference between two cables that are identical besides the cone orientation. A number of factors, for instance cable type, size, and the number of optical cores, may cause this temperature difference to vary. Understanding these factors will be critical to developing a robust, empirical relationship between the

directionally-sensitive temperature signals and wind direction. Future work will need to be guided be these questions and explore the remaining potential dependencies.





## 4   Conclusions

By combining fiber-optic distributed sensing techniques with independent thermal infrared imagery and computational fluid flow simulations we evaluated and verified a method for detecting distributed wind direction using microstructures affixed to actively heated fiber optic cables. We demonstrated that the microstructures, which are affixed to a pair of fiber-optic cables

in opposing directions, introduce a directional sensitivity of the turbulent heat loss from the cable to the air. This differential convective heat flux can be detected as a temperature difference between the two cables. The temperature difference then allows for computing wind direction along the axis of the fiber, providing a method for observing wind direction on a distributed basis. The work presented here thus represents a critical step in employing the microstructure approach to achieve the ultimate goal of building a spatially-resolving, three-dimensional flow sensor for the atmospheric surface layer to record turbulent fluxes of

sensible heat and momentum.

*Author contributions.*   John Selker and Christoph Thomas formulated the original concept for this study. All coauthors developed the experimental design. Johann Schneider built the wind tunnel with further contributions from all coauthors. Karl Lapo, Anita Freundorfer, and Lena Pfister performed the analysis and experiments. Karl Lapo prepared the manuscript with contributions from all coauthors.

*Acknowledgements.*   This project has received funding from the European Research Council (ERC) under the European Union's Horizon

2020 research and innovation programme under grant agreement No 724629, project DarkMix. Initial prototyping and testing of the microstructure approach was conducted at Oregon State University, Corvallis, Oregon, at the Openly Published Environmental Sensing Lab (open-sensing.org), with support from Cara Walter, and at the Experimental Fluid Mechanics Research Lab with support from Dr. James Liburdy. Dr. Chad Higgins provided insights during the original development and testing of the microstructure concept. Further assistance in operating the fiber-optic sensing system in the OSU wind tunnel was provided by Justus van Ramshorst.



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
