# Peer review of "Distributed observations of wind direction using microstructures attached to actively heated fiber-optic cables"

_Atmospheric Measurement Techniques, 2019_

## Referee Comment (RC1) · Anonymous Referee #2 · 23 Dec 2019

This study measures temperature differences between two parallel actively heated fiber-optic cables with microstructures to further determine the wind direction. The study fits the scope of *Atmospheric Measurement Techniques*. The authors have addressed most of my previous comments. However, I still have one major concern.

Major comments

(1) There are unclear places in the derivation of equation (6) from equation (3).

Equation (3)

$$Q_h = \rho u_*^{1-m} c_p (T_a - T_s) \alpha^{-1} \left(\frac{h}{\nu}\right)^{-m} Pr^{-n}$$

The above equation leads to

$$Q_{left} - Q_{right} = \rho u_*^{1-m} c_p \left(\frac{h}{\nu}\right)^{-m} Pr^{-n} \left[\frac{(T_a - T_{left})}{\alpha_{left}} - \frac{(T_a - T_{right})}{\alpha_{right}}\right]$$

Assuming $Pr = 1$, the above equation is reduced to

$$\frac{Q_{left} - Q_{right}}{\rho c_p \left(\frac{h}{\nu}\right)^{-m}} u_*^{m-1} = \frac{T_a}{\alpha_{left}} - \frac{T_{left}}{\alpha_{left}} - \frac{T_a}{\alpha_{right}} + \frac{T_{right}}{\alpha_{right}} = \frac{T_{right}}{\alpha_{right}} - \frac{T_{left}}{\alpha_{left}} + \left(\frac{1}{\alpha_{left}} - \frac{1}{\alpha_{right}}\right) T_a$$

Assuming $\alpha_{left} \ll \alpha_{right}$, i.e., $\frac{1}{\alpha_{left}} \gg \frac{1}{\alpha_{right}}$ if $\alpha > 0$, the above equation is reduced to

$$\frac{Q_{left} - Q_{right}}{\rho c_p \left(\frac{h}{\nu}\right)^{-m}} u_*^{m-1} = \frac{T_{right}}{\alpha_{right}} - \frac{T_{left}}{\alpha_{left}} + \frac{T_a}{\alpha_{left}}$$

However, $\alpha_{left} \ll \alpha_{right}$ does not lead to $\frac{\alpha_{right}}{\alpha_{left}} \approx \frac{1}{\alpha_{left}}$ (this is the assumption in the author's response), unless $\alpha_{right} \approx 1$ is assumed.

Even if $\alpha_{right} \approx 1$ is assumed, the above equation is reduced to

$$\frac{Q_{left} - Q_{right}}{\rho c_p \left(\frac{h}{\nu}\right)^{-m}} u_*^{m-1} = T_{right} - \frac{T_{left}}{\alpha_{left}} + \frac{T_a}{\alpha_{left}}$$

Please show how to derive from the above equation to equation (6), i.e.,

$$\frac{Q_{left} - Q_{right}}{\rho c_p \left(\frac{h}{\nu}\right)^{-m}} u_*^{m-1} = \frac{1}{\alpha_{left}} \frac{1}{\alpha_{right}} (T_{left} - T_{right})$$

---

## Referee Comment (RC2) · Anonymous Referee #1 · 26 Dec 2019

Review AMT-2019-188

DEC19

General remarks The paper introduces an interesting concept to measure wind direction with Fiber Optics Distributed Sensing (FODS). The paper is well written. The experiments are well described and the results are carefully worded. This proof of concept is interesting

Major remarks There are no serious flaws in the paper, as far as I can discern. The only thing I would like to see some remark is the effect of buoyancy at (very?) low windspeeds. The heated cable will set up its own convection at low windspeeds. It

should not be difficult to say something about this. Has this been taken into account by the OpenFOAM simulation? Normally, the effect would be small due to the small diameter of the cable but with the cones, the effective diameter may be large, especially when the cones are narrowly spaced at 2cm.

Minor remarks P1 L20: Petrides et al (doi:10.1029/2010WR009482) is probably the earliest published atmospheric application of FODS. P3 L2: First use of FOC, please write out acronym. Fig1: Where in the tunnel was the sonic anemometer placed? P3 L15: What is U? P4 L17: 'to' missing after 'used' P4 L21: Does everyone know what 'turbulent intensity' means? P4 L26: Perhaps it is stated somewhere but please state here (and in caption Fig2) over how long the windspeed is being averaged. P4 L29: What one really would like to know is how well the cable is captured at this resolution. With field of view and distance from cable, this is easy to calculate. P5 L15: 0.127m is the sampling resolution. The actual resolution is about 0.27m. P7 L6: Capital delta. P14 L8: Dangling modifier: Who/what reviews?

---

## Short Comment (SC1) · 14 Jan 2020

The manuscript provides a very nice overview of the proof-of-concept of the wind direction measurement using DTS. It serves as a good basis for further studies and future application of this method in the field. I only have a few small questions/comments for clarification, mostly related to the DTS device/method.

General comments:

Was a longer time average of the DTS data used, or only the 1s resolution for the analysis? I did not see this clearly mentioned in section 2.2. It does come back in section

3.3, but perhaps it could be expanded upon earlier, to make the relation between timeaveraging and uncertainty more clear for the reader (i.e., measurement uncertainty which decreases with the square root of the amount of samples).

Is an estimate available for the response time of the FO cables (with the attached cones) used in this study? If the response time is (much) slower than the 1 second averaging time, it could be more logical to average over a longer time.

Specific comments:

Page 5, line 25; Why does the DTS device have a temperature resolution of 0.01 K? The data resolution of the Stokes/anti-Stokes data (6 significant figures) results in a resolution of 0.001 K. It might be more clear to state the expected noise level of the device at a certain integration time.

Page 5, line 26; The spatial resolution of Silixa's Ultima devices is 30-35 cm, sampled at an interval of 12.7 cm.

Page 5, line 28; I assume that a single-ended calibration is used?

Page 6, line 1; The RMSE of the bath is mentioned, but not the bias. I assume the mean bias in your reference bath is really low, so it could be good to make a distinction between the measurement noise/uncertainty and bias.

Page 11, figure 4; the unlabelled y-axis of figure 4d is not aligned with figure fc

Page 16, line 20; Coenders-Gerrits is with capital G.

---

## Author Comment (AC1) · 3 Feb 2020

This study measures temperature differences between two parallel actively heated fiber-optic cables with microstructures to further determine the wind direction. The study fits the scope of Atmospheric Measurement Techniques. The authors have addressed most of my previous comments. However, I still have one major concern.

Thank you for your comments. They strengthened the paper.

5    Major comments (1) There are unclear places in the derivation of equation (6) from equation (3).

Here I omit the equations for brevity

Please show how to derive from the above equation to equation (6), i.e.,

$$\frac{Q_{left} - Q_{right}}{\rho c_p \left(\frac{h}{\nu}\right)^{-m}} u_*^{(m-1)} = \left(\frac{(T_a - T_{right})}{\alpha_{right}} - \frac{(T_a - T_{left})}{\alpha_{left}}\right)$$

You are correct, we made a mistake in the algebra getting from equation 3 to equation 6. We now have revised section 2.1.

10    Equation 6 is now:

$$\left(\frac{T_{left}}{\alpha_{left}} - \frac{T_{right}}{\alpha_{right}}\right) = \frac{Q_{right} - Q_{left}}{\rho c_p (\frac{h}{\nu})^{-m}} u_*^{m-1} - T_a \left(\frac{1}{\alpha_{right}} - \frac{1}{\alpha_{left}}\right)$$

and the text has been revised to make it explicit that we are only searching for a non-linear decaying relationship between wind speed and the temperature difference, not an exact representation.

---

## Author Comment (AC2) · 3 Feb 2020

Response to RC2.

General remarks The paper introduces an interesting concept to measure wind direction with Fiber Optics Distributed Sensing (FODS). The paper is well written. The experiments are well described and the results are carefully worded. This proof of concept is interesting

Thank you for your comments and review. We hope that we could adequately address them and strengthen the manuscript.

Major remarks

There are no serious flaws in the paper, as far as I can discern. The only thing I would like to see some remark is the effect of buoyancy at (very?) low windspeeds. The heated cable will set up its own convection at low windspeeds. It should not be difficult to say something about this. Has this been taken into account by the OpenFOAM simulation? Normally, the effect would be small due to the small diameter of the cable but with the cones, the effective diameter may be large, especially when the cones are narrowly spaced at 2cm.

We used a solver in OpenFOAM that does not include heat transfer, which is why we only show turbulent kinetic energy in Figure 5d-f as a proxy for the sensible heat flux. Part of our motivation is that the approach only requires the temperature difference between fibers, not the absolute value of their temperatures. The coned fibers were always relatively close to each other in temperature (e.g. < 3.5K) so the difference in the buoyancy driven flux of heat should not be significant. As the wind speed approaches zero, the temperature difference between the fibers should also shrink as the difference in the convective heat flux away from each fiber should disappear. We noted this behavior when the wind tunnel was off. If the temperature difference is close to 0K than the difference in the buoyancy heat flux between the fibers should be vanishingly small. Further, a numerical study using similar triangular shapes but with much larger temperature differences demonstrates no discernible buoyancy effects, further justifying neglecting this term. We remark on these details in section 2.4.

Minor remarks

I think there may be a confusion on which version was reviewed. The page/line numbers indicated line up with the initial submission and not the corrected submission posted for discussion. We hope that we were able to correctly address your comments.

P1 L20: Petrides et al (doi:10.1029/2010WR009482) is probably the earliest published atmospheric application of FODS.
Citation added

P3 L2: First use of FOC, please write out acronym.
Fixed.

Fig1: Where in the tunnel was the sonic anemometer placed?
We clarify in section 2.2 that the sonic anemometer was only used to characterize the wind tunnel. The CSAT is large enough to create flow distortions near itself that affected are initial tests using this device. This is why we opt to use the one-dimension hot wire anemometer for the tests with the fiber, as depicted in Figure 1.

P3 L15: What is U?
U should now be defined.

P4 L17: "to" missing after "used"
Fixed.

P4 L21: Does everyone know what "turbulent intensity" means?
Added a short description to turbulent intensity and a citation to a study describing turbulent intensity for a variety of atmospheric conditions.

P4 L26: Perhaps it is stated somewhere but please state here (and in caption Fig2) over how long the windspeed is being averaged.

Added the time scale to this part of the text as well as in Figure 2.

P4 L29: What one really would like to know is how well the cable is captured at this resolution. With field of view and distance from cable, this is easy to calculate.

Each pixel is approximately 0.24mm in width/length so each fiber has a little over 4 pixels width. From this we arrived at our decision to use the three warmest pixels for each y location, as this choice assures that we only sample fiber pixels.

P5 L15: 0.127m is the sampling resolution. The actual resolution is about 0.27m.

Fixed.

P7 L6: Capital delta.

Fixed.

P14 L8: Dangling modifier: Who/what reviews?

Fixed in the general revision to the scope of the discussion.

---

## Author Comment (AC3) · 3 Feb 2020

The manuscript provides a very nice overview of the proof-of-concept of the wind direction measurement using DTS. It serves as a good basis for further studies and future application of this method in the field. I only have a few small questions/comments for clarification, mostly related to the DTS device/method.

Thank you for the comments! I hope I have responded to them in a satisfactory manner.

5

General comments: Was a longer time average of the DTS data used, or only the 1s resolution for the analysis? I did not see this clearly mentioned in section 2.2. It does come back in section but perhaps it could be expanded upon earlier, to make the relation between time averaging and uncertainty more clear for the reader (i.e., measurement uncertainty which decreases with the square root of the amount of samples).

- 10 While the Ultima is capable of very high temperature precision, these results can only be achieved over long time averages. At shorter time scales the DTS has substantial noise. As we want to use the shortest time integration possible for future deployments, we explored how the noise at the finest resolution interacts/interferes with our ability to discern the wind direction signal in the temperature difference. We used a ten-minute sampling interval as it was long enough to characterize the noise for a given experiment, but short enough that we could do a large number of experiments.
- 15

35

45

Is an estimate available for the response time of the FO cables (with the attached cones) used in this study? If the response time is (much) slower than the 1 second averaging time, it could be more logical to average over a longer time.

We do not address the response time of the cone/fiber/DTS system in this study due to the artifacts from using a wind tunnel.
We only recorded data after the fiber reached an equilibrium temperature, which would take on the order of minutes. From previous studies with DTS, the time lag is not realistic behavior in an atmospheric deployment (wind speed time response is on the order of 10s, air temperature response is less than that). Based on observing the IR camera at the time, we suspect that we were measuring the time response of the entire wind tunnel and not just the fiber. To not leave you completely dissatisfied, initial tests with our wind direction approach in an environmental deployment shows a time lag between 10-15 seconds with the exact value depending on the meteorological conditions. These results are the subject of an upcoming publication that we

25 are very excited for.

Specific comments: Page 5, line 25; Why does the DTS device have a temperature resolution of 0.01 K? The data resolution of the Stokes/anti-Stokes data (6 significant figures) results in a resolution of 0.001 K. It might be more clear to state the expected noise level of the device at a certain integration time.

30 Following our conversation on the topic, the relevant number here for this manuscript is the instrument noise at the finest temporal and spatial resolution. We removed the reference to the instrument resolution and streamlined the introduction of the DTS device as a result.

Page 5, line 26; The spatial resolution of Silixa's Ultima devices is 30-35 cm, sampled at an interval of 12.7 cm. Number updated to 0.127m.

Page 5, line 28; I assume that a single-ended calibration is used? We explicitly state this now. Thank you!

40 Page 6, line 1; The RMSE of the bath is mentioned, but not the bias. I assume the mean bias in your reference bath is really low, so it could be good to make a distinction between the measurement noise/uncertainty and bias. We explicitly ignore the bias as we only compare the instrument to itself through the temperature difference.

Page 11, figure 4; the unlabelled y-axis of figure 4d is not aligned with figure c Good catch. Fixed.

Page 16, line 20; Coenders-Gerrits is with capital G. Fixed.